# Dissociative Effects of Neuropeptide S Receptor Deficiency and Nasal Neuropeptide S Administration on T-Maze Discrimination and Reversal Learning

**DOI:** 10.3390/ph14070643

**Published:** 2021-07-05

**Authors:** Ahmet Oguzhan Bicakci, Pei-Ling Tsai, Evelyn Kahl, Dana Mayer, Markus Fendt

**Affiliations:** 1Institute for Pharmacology and Toxicology, Medical Faculty, Otto-von-Guericke University Magdeburg, 39120 Magdeburg, Germany; ahmetoguzhanbicakci@gmail.com (A.O.B.); pei-ling.tsai@st.ovgu.de (P.-L.T.); evelyn.kahl@med.ovgu.de (E.K.); dana.mayer@med.ovgu.de (D.M.); 2Integrative Neuroscience Program, Otto-von-Guericke University Magdeburg, 39120 Magdeburg, Germany; 3Center for Behavioral Brain Sciences, Otto-von-Guericke University Magdeburg, 39120 Magdeburg, Germany

**Keywords:** discrimination learning, learning strategy, nasal administration, neuropeptide S (NPS), neuropeptide S receptor (NPSR), reversal learning, T-maze

## Abstract

Cognitive flexibility refers to the ability to modify learned behavior in response to changes in the environment. In laboratory rodents, cognitive flexibility can be assessed in reversal learning, i.e., the change of contingencies, for example in T-maze discrimination learning. The present study investigated the role of the neuropeptide S (NPS) system in cognitive flexibility. In the first experiment, mice deficient of NPS receptors (NPSR) were tested in T-maze discrimination and reversal learning. In the second experiment, C57BL/6J mice were tested in the T-maze after nasal administration of NPS. Finally, the effect of nasal NPS on locomotor activity was evaluated. NPSR deficiency positively affected the acquisition of T-maze discrimination but had no effects on reversal learning. Nasal NPS administration facilitated reversal learning and supported an allocentric learning strategy without affecting acquisition of the task or locomotor activity. Taken together, the present data show that the NPS system is able to modulate both acquisition of T-maze discrimination and its reversal learning. However, NPSR deficiency only improved discrimination learning, while nasal NPS administration only improved reversal learning, i.e., cognitive flexibility. These effects, which at first glance appear to be contradictory, could be due to the different roles of the NPS system in the brain regions that are important for learning and cognitive flexibility.

## 1. Introduction

Cognitive flexibility is an important executive function which describes the cognitive processes underlying adaptive behavioral changes in response to changes in the environment [1,2]. This includes reversal learning as well as the transfer of acquired knowledge across different stimulus categories or task characteristics [3]. An example of reversal learning is when contingencies in a discriminative task change, i.e., when a behavior that was previously rewarded is now not rewarded anymore (e.g., going left in a T-maze), and behavior that was previously not rewarded is now rewarded (e.g., going right in a T-maze). Notably, cognitive flexibility is impaired in a number of human disorders, including several neurodegenerative and neuropsychiatric disorders, such as Alzheimer’s disease or schizophrenia [4,5]. Animal and clinical studies have revealed that cognitive flexibility is associated with different subregions of the frontal cortex [6,7,8,9]. Dysfunctions within these subregions lead to impaired cognitive flexibility in humans and rodents [7,10,11,12,13]. Furthermore, the frontal cortex receives input from the limbic system, which is also involved in the mediation and modulation of cognitive flexibility, as has been shown for the hippocampus [14,15,16] and the amygdala [17,18].

While several studies have shown that some of the classical neurotransmitters play an important role in cognitive flexibility [12,19,20,21], the role of neuropeptides in cognitive flexibility is poorly understood. This is surprising, since terminals and receptors of different neuropeptides are (1) localized within the regions of the frontal cortex involved in cognitive flexibility, and (2) down-regulated in neuropsychiatric disorders which are associated with deficits in cognitive flexibility [22]. One exception is probably the neuropeptide orexin, whose levels in the cerebrospinal fluid are upregulated in those schizophrenia patients with fewer negative and disorganized symptoms [23], and for which there is some research regarding its role in cognitive flexibility [24,25].

The neuropeptide S (NPS) system was discovered and described about 20 years ago [26]. Many studies have revealed that the NPS system is involved in a variety of brain functions ranging from arousal via fear and anxiety to reward-related processes [26,27,28,29,30,31,32,33,34]. In addition, the NPS system also plays a role in learning and memory processes. For example, NPS administration enhances inhibitory avoidance learning [35], while NPS deficiency reduces inhibitory avoidance learning [36]. NPS administration has memory-enhancing effects in novel object recognition and Morris water maze [37,38,39,40] and is able to rescue memory deficits in these and other tests [38,39,41]. So far, the question of whether the NPS system is involved in cognitive flexibility has not been addressed. However, the NPS system would be well suited to modulating cognitive flexibility, since NPS receptors (NPSR) are highly expressed in some brain areas critical for cognitive flexibility [42,43] and NPSR signaling can increase neuronal excitability in these areas [44].

The aim of the present study was to investigate a potential role of the NPS system in cognitive flexibility. To measure cognitive flexibility, we utilized T-maze discrimination and reversal learning [45]. Three experiments were performed: NPSR-deficient mice were tested in T-maze discrimination and reversal learning; the effects of nasal NPS administration on T-maze discrimination and reversal learning were investigated in C57BL/6J mice; and potential confounding effects of nasal NPS administration on locomotor activity were tested in the open field. We found that NPSR deficiency positively affected the acquisition of T-maze discrimination but had no effects on reversal learning, while nasal NPS administration did not affect acquisition of T-maze discrimination but facilitated reversal learning without affecting locomotor activity.

## 2. Results

### 2.1. Effects of NPSR Deficiency on T-maze Discrimination and Reversal Learning

NPSR-deficient mice and their heterozygous and wild-type littermates (group sizes: NPSR^+/+^ females: 6, males: 7; NPSR^+/−^ females: 6, males: 8; NPSR^−/−^ females: 6, males: 9) were submitted to a T-maze discrimination and reversal learning task. Briefly, the mice had to learn during the acquisition phase that only one of the short arms of the T-maze was rewarded with a sucrose pellet. Ten trials per day on five consecutive days (A1–A5) were performed. After a break of two days, the contingencies were changed, i.e., now the other short arm of the T-maze was rewarded. Again, ten trials per day on five consecutive days (reversal phase: R1–R5) were performed. Furthermore, on days A5 and R5, an additional trial was performed to determine the learning strategy of the mice. For this trial, the T-maze was turned by 180 degrees (the environmental cues stayed at the same place) and both arms were rewarded. When the mouse chose the arm with the same spatial location as before, the strategy was considered to be an allocentric (contextual/spatial) one, while choosing the arm with the same direction (left/right) as before was considered to be an egocentric (response) strategy [46].

Figure 1a depicts the number of correct choices in the acquisition phase. An ANOVA using genotype and sex as between-subject factors and days (A1–A5) as within-subject factor revealed a significant main effect of days (F_4,140_ = 24.67, *p* < 0.0001) indicating successful acquisition of T-maze discrimination learning. In general, male mice showed more correct responses as indicated by a main effect of sex (F_1,35_ = 5.04, *p* = 0.03). There was neither a main effect of genotype nor significant interactions between two or three of these factors (F-values < 1.72, *p*-values > 0.10). However, post-hoc comparisons revealed that NPSR^−/−^ mice showed better performance on day A5 than NPSR^+/+^ mice (Dunnett’s multiple comparison test: q = 2.53, *p* = 0.03; Figure 1a). Since reversal learning is only possible in mice that acquired T-maze discrimination [47], we only used “learners” for the analyses of reversal learning and learning strategy and excluded “non-learners”. Learners were defined by having increased their number of correct responses from A1 to A5 by more than one response. Notably, the proportion of non-learners was significantly lower in NPSR^−/−^ mice (1 of 15) than in NPSR^+/+^ mice (5 of 13), as revealed by chi-square test (χ2 = 4.54, *p* = 0.03). The proportions of non-learners were no different in NPSR^+/+^ and NPSR^+/−^ mice (6 of 14; χ2 = 0.05, *p* = 0.82). Furthermore, there were no sex differences (females: 6 of 18, males: 6 of 24; χ2 = 0.23, *p* = 0.63).

After excluding the non-learners, there was a main effect of days (Figure 1b; F_4,96_ = 36.89, *p* < 0.0001) and a trend for a better performance for male mice in the acquisition phase (F_1,24_ = 3.88, *p* = 0.06; see also Figure 1f,j). The other factors and interactions did not reach statistical significance (F-values < 1.06, *p*-values > 0.37). Very similar results were seen in the reversal phase (R1–R5). There was a main effect of days (F_4,96_ = 75.56, *p* < 0.0001) suggesting successful reversal learning, but other factors and interactions did not reach statistical significance (F-values < 2.84, *p*-values > 0.10). This is supported by the analysis of the switch cost ratio, i.e., the ratio of mean performance during acquisition (A1–A5) and the mean performance during reversal learning (R1–R5), showing no effects of genotype (Figure 1c; F_2,26_ = 1.05, *p* = 0.37). Furthermore, there were no differences in the learning strategy (Figure 1d; allocentric vs. egocentric learning), either in the acquisition phase (χ2 = 3.53, *p* = 0.17) or in the reversal phase (χ2 = 0.37, *p* = 0.83).

Figure 1e–l show the data separated for female and male mice. The pattern of the results is basically the same as for the pooled data except that in female mice, a significant interaction of time and genotype was found (F_8,60_ = 2.23, *p* = 0.03). Post-hoc comparisons revealed that this interaction was based on the fact that female NPSR^+/−^ mice did not learn the discriminative task (Dunnett’s multiple comparison test, comparison A1 vs. A5: q = 0.49, *p* = 0.96; Figure 1e). Furthermore, the comparisons of the performance of NPSR^+/+^ and NPSR^−/−^ mice on day A5 did not reach statistical significance, if separately analyzed in the two sexes (females: q = 2.09, *p* = 0.32; males: q = 1.98, *p* = 0.07), but suggest that the effects seen in the pooled data are driven by the male mice. Of note, group sizes are small in these separated analyses for the two sexes, i.e., type I errors cannot be excluded.

### 2.2. Effects of Nasal NPS Administration on T-maze Discrimination and Reversal Learning

To further investigate the role of the NPS system in T-maze discrimination and reversal learning, we applied NPS nasally to male C57BL/6J mice. We only used male mice in this experiment, since they had shown a more robust and less variable performance in the previous experiment. NPS (10 µL of 1 mM solution) was administered nasally daily, 2.5 h before the 10 trials. Administration dose, volume, and time were chosen based on literature data [40,48] and our own pilot studies.

Forty mice were submitted to T-maze discrimination learning, half of them after nasal administration of saline (vehicle), half of them after nasal NPS administration (Figure 2a). An ANOVA with treatment as between-subject factor and days (A1–A5) as within-subject factor revealed a significant main effect of days (F_4,152_ = 35.93, *p* < 0.0001), indicating successful acquisition of T-maze discrimination learning. However, there was neither a main effect of nasal NPS administration (F_1,38_ = 0.36, *p* = 0.55) nor an interaction between treatment and days (F_4,152_ = 0.89, *p* = 0.47), showing that NPS did not affect the acquisition of T-maze discrimination learning. As in the previous experiment, the non-learners were excluded for the analyses of reversal learning and learning strategy (5 mice/treatment). In the mice that acquired T-maze discrimination learning (F_4,112_ = 54.30, *p* < 0.0001; treatment: F_1,28_ = 1.05, 0.32; interaction: F_4,112_ = 0.96, *p* = 0.43), successful reversal learning was observed (Figure 2b; days R1–R5: F_4,112_ = 86.82, *p* < 0.0001). Nasal administration of NPS improved reversal learning, as indicated by a main treatment effect (F_1,28_ = 5.82, *p* = 0.02), however, there was no treatment x days interaction (F_4,112_ = 0.12, *p* = 0.98). This positive effect of nasal NPS administration learning was confirmed by the analysis of the switch cost ratio, which was significantly reduced in NPS-treated mice (Figure 2c; *t*-test: t_28_ = 2.13, *p* = 0.04). Notably, NPS-treated mice also changed the learning strategy from an egocentric one in the acquisition phase to an allocentric one in the reversal phase (Figure 2d; χ2 = 5.00, *p* = 0.03), while vehicle-treated mice did not change the learning strategy (χ2 = 0.14, *p* = 0.70). There were no effects of NPS administration on strategy choice during the acquisition or reversal phase (χ2 < 2.15, *p* < 0.14).

### 2.3. Effects of Nasal NPS Administration on Locomotor Activity

We then evaluated whether nasal NPS administration affected locomotor activity. Male C57BL/6J and NPSR^+/+^ mice (*n* = 8 and 6, respectively, i.e., total *n* = 14) were placed into an open field and locomotor activity was measured for one hour. Vehicle or NPS was then nasally administered (balanced across animals), the mice were put back into the open field, and locomotor activity was measured for a further three hours. The mice were tested again on the following day with the reverse treatment to that used previously.

Figure 3 depicts the horizontal locomotor activity (distance travelled in arbitrary units) in the open field, before and after nasal treatment. An ANOVA using mouse line (C57BL/6J and NPSR^+/+^) as between-subject factor and phase (before/after) and treatment (vehicle, NPS) as within-subject factors revealed a main effect of phase (F_1,12_ = 103.83, *p* < 0.0001) but not of mouse line or treatment (F-values < 0.81, *p*-values > 0.38). Importantly, there was no interaction between phase and treatment (F_1,12_ = 0.08, *p* = 0.79), indicating that nasal NPS administration did not affect locomotor activity.

## 3. Discussion

The aim of the present study was to test the role of the NPS system in T-maze discrimination and reversal learning. We found that NPSR deficiency improved the acquisition of t T-maze discrimination learning but did not affect its reversal learning. Nasal administration of NPS facilitated reversal learning without affecting the acquisition of the task. Furthermore, nasal NPS administration had no effect on locomotor activity.

The scientific rationale of our study was based on literature findings demonstrating that NPS administration had beneficial effects on learning and memory [35,37,38,39,40,41]. However, the effect of NPS administration on cognitive flexibility has not previously been tested. We used discrimination learning in the T-maze, since this task is relatively well learned by mice [49] and its reversal is a measure of cognitive flexibility [50]. In our first experiment, NPSR-deficient mice were tested in the T-maze. These mice are a commonly used tool to investigate functions of the NPS system [29,51,52], because there is a well-investigated human polymorphism of the NPSR gene that is associated with an increased risk for panic disorders [53,54,55,56]. However, NPSR-deficient mice are relatively inconspicuous in most behavioral tests and only show modest, if any, changes, such as being slightly less active, more anxious, more fearful or more stress-sensitive than their wild-type littermates [29,31,51,52,57,58,59]. Novel object recognition, as well as the learning and extinction of classical conditioned fear, is not affected by NPSR deficiency [29,59], while safety learning is improved in male NPSR-deficient mice [52]. Furthermore, the NPSR genotype did not affect the acquisition and expression of conditioned social fear, but its extinction was impaired in heterozygous and facilitated in homozygous NPSR-deficient mice [58]. In our study, NPSR deficiency positively affected the acquisition of T-maze discrimination learning, since NPSR^−/−^ mice showed higher performance at the end of the acquisition phase (Figure 1a), and the rate for non-learners was significantly lower. Furthermore, male mice performed significantly better than female mice, but there were no interactions with the factors genotype and days. For the analysis of reversal learning, we excluded the mice that did not acquire the task and noted that there were significantly less non-learners in the NPSR^−/−^ group. However, neither reversal learning and switch cost ratio nor the learning strategy of the mice were then affected by NPSR deficiency (Figure 1b–d). This general pattern of the results was confirmed by separated analyses of the sexes, which also revealed that female NPSR^+/−^ mice did not acquire the task (Figure 1e). Taken together, homozygous NPSR deficiency improved the performance in T-maze discrimination (higher performance on day A5 and lower rate of non-learners) but had no effects on reversal learning. This indicates beneficial effects of NPSR deficiency. The beneficial effects of NPSR deficiency on learning were also observed in a previous study we undertook, in which male NPSR-deficient mice showed a better performance in safety learning [52], while other studies have reported no effects of NPSR deficiency on learning [29,59]. The observed performance differences could be based on the NPSR deficiency itself, but may also be due to compensatory mechanisms, e.g., up- or down-regulation of other transmitter systems [60].

In our second experiment, we tested the effects of nasal NPS administration on T-maze discrimination and reversal learning in C57BL/6J mice. We chose nasal NPS administration, since previous studies showed that nasally applied NPS reaches the brain and induces behavioral changes [40,48]. Notably, these behavioral changes include anxiolytic-like effects and an improvement of novel object recognition but no increase in locomotor activity. The latter is very pronounced after NPS administration into the cerebral ventricles [26,30,61] and could interfere with T-maze performance. We performed this experiment in male C57BL/6J mice, based on the observation in our first experiment that male mice showed a more robust performance in T-maze learning, and the NPSR genotype effects were more pronounced in male mice. However, in the present study, the acquisition of T-maze discrimination learning was not affected by nasal NPS administration (Figure 2a). Again, we excluded the non-learners whose rates did not differ across treatments. Further analyses showed that nasally applied NPS improved reversal learning (Figure 2b). This NPS effect was weak but statistically significant, and was confirmed by a significantly lower switch cost ratio (Figure 2c). Furthermore, nasal NPS changed the learning strategy in the reversal phase towards an allocentric (i.e., contextual) strategy. When rodents acquire a discrimination task in a T-maze they usually prefer an allocentric learning strategy at the beginning, and with more experience they then switch to an egocentric strategy [62]. Strategy choice is also notably affected by the availability of contextual stimuli. To the best of our knowledge, it is not known how reversal learning affects strategy choice in the T-maze. The present data show that—in our laboratory—most control-treated mice preferred an egocentric strategy throughout the experiment, while NPS-treated mice changed to an allocentric strategy during the reversal phase. An explanation for this observation could be that NPS treatment increases excitability of the hippocampus, the brain region involved in allocentric learning [62]. This may support choosing an allocentric strategy, as well as reversal learning [63]. In an additional experiment, we confirmed previous findings that nasal NPS administration does not affect locomotor activity (Figure 3). Taken together, nasal NPS administration improved reversal learning and induced a shift to an allocentric learning strategy without affecting the acquisition of the task or locomotor activity.

We hypothesize that the pharmacokinetics of nasal NPS administration play a role in this combination of effects and no-effects. Ionescu and colleagues investigated the distribution of NPS in the brain after nasal administration in comparison to intracerebroventricular administration [48]. Brain areas that are not well reached by nasal administration include cortical areas, the basal ganglia, most amygdala subregions, the locus coeruleus and adjacent nuclei, as well as the cerebellum; while the forebrain, hippocampus, thalamus, hypothalamus and most midbrain and brainstem areas are equally well reached by nasal and intracerebroventricular administrations [48]. Acquisition of discrimination learning in the T-maze is mainly learned with an egocentric strategy, which involves the basal ganglia [62]. Furthermore, the basal ganglia are involved in regulating and modulating locomotor activity [64]. NPSR are highly expressed within the basal ganglia [42,43,65]; however, nasally applied NPS does not reach the basal ganglia even at high concentrations [48], which substantiates our finding that nasally applied NPS does not affect the acquisition of T-maze discrimination learning or locomotor activity. In contrast, nasally applied NPS reaches the hippocampus and most probably—at least this was shown for other neuropeptides [66]—also the prefrontal cortex. The hippocampus is involved in allocentric, i.e., contextual learning strategies [62], while parts of the frontal cortex are involved in reversal learning [50]. Nasal NPS administration should stimulate or increase the excitability of neurons in these brain areas (these NPS effects have been shown in the amygdala [33,44,67,68]) and thereby facilitate reversal learning as well as the use of an allocentric strategy. We can confirm this hypothesis with the present study. Importantly, the present findings do not exclude the possibility of NPS having an effect on the acquisition of T-maze discrimination learning if it were administered in a way that leads to high NPS concentrations in the basal ganglia (e.g., via local or intracerebroventricular injections). However, we would then also expect pronounced effects on locomotor activity [69].

In summary, the present study demonstrates that NPSR deficiency leads to an improved acquisition of a discriminative task in the T-maze without affecting its reversal learning. In contrast, nasal administration of NPS facilitates T-maze reversal learning and the use of an allocentric learning strategy without affecting the acquisition of the task or locomotor activity. This suggests that an impaired NPS system (NPSR deficiency) is beneficial for acquisition, while a facilitated NPS system (nasal NPS administration) is beneficial for reversal learning. Future studies should investigate the physiological and/or neuroanatomical underpinnings of these different effects. We hypothesize that the NPS system plays different roles in brain systems involved in discriminative learning and cognitive flexibility. With regard to nasal NPS administration, it would also be of interest whether the observed effects can also be seen in other mouse measures of cognitive flexibility as well as in mouse models of diseases associated with impaired cognitive flexibility. Furthermore, the specificity of the observed effects could be evaluated by testing nasal NPS administration in NPSR-deficient mice and/or mice pre-treated with NPSR antagonists.

## 4. Materials and Methods

### 4.1. Animals

Two- to three-month-old experimentally naive male C57BL/6J mice (*n* = 40), and male and female wild-type, heterozygous, and homozygous NPSR-deficient mice (B6.129S6/SvEvTac-Npsr1^tm1Bhk^ [70], *n* = 42) were used. All mice were bred in the institute’s animal facility and housed in groups of 4–8 per cage under temperature- and humidity- controlled standard conditions and a 12 h light/dark cycle with lights on at 6:00 am. The behavioral experiments were conducted between 9:00 am and 5:00 pm. Water and food were available ad libitum, except for the two days before and during the T-maze experiments, where mice were restricted to an amount of food (2–2.5 g/mice/day) to maintain approximately 90% of their basal body weight. All experiments were performed in agreement with the European regulations for animal care and their use for experiments (2010/63/EU) and were approved by the local authorities (Landesverwaltungsamt Sachsen-Anhalt, Az. 42502-3-747 approved on 21 April 2015 and Az. 42502-2-1351 UniMD approved on 15 December 2015).

### 4.2. Drug Administration

NPS (Cat. No. 5857; Tocris/Bio-Techne GmbH, Wiesbaden, Germany) was dissolved in saline. For nasal administration, 10 µL of 1 mM NPS solution (i.e., 10 nmol) or vehicle were applied topically on the rhinarium area of the awake and scruff-fixed mice, distributed equally on both sides using a micropipette [40].

### 4.3. Food Reward

As a food reward, 14-mg non-flavored sucrose rodent tablets were used (5TUT, #1811250; TestDiet, Richmond, IN, USA).

### 4.4. T-Maze

A custom-made T-maze with a gray floor (polyvinyl chloride) and transparent sidewalls (acryl glass) was used (long arm: 109 cm; short arms: 61 cm; width and height of the arms: 12 cm). The sucrose pellets were placed in small 0.5 cm deep wells on the end of the left and right short arm. Guillotine doors separated a start section from the rest of the long arm, and separated the short arms from the long arm. The T-maze was located in a laboratory room with several obvious spatial cues.

### 4.5. Open Field

An arena with dark walls (45 cm × 45 cm × 30 cm) surrounded by a frame with infrared light sensors (14 mm distance) was used. A software (TSE Fear Conditioning System; TSE Systems, Bad Homburg, Germany) used the signals from these sensors to measure locomotor activity.

### 4.6. Behavioral Procedure (T-Maze)

In the week before the start of the T-maze experiment, the animals were handled (5 min on two days) and habituated to the T-maze (group and single habituation, each 5 min). From two days before the T-maze experiments, the mice were food restricted. The actual T-maze experiment consisted of two phases: an acquisition phase and a reversal phase. Both phases lasted five days (A1–A5, R1–R5) with 10 trials daily, with a break of two days between the two phases. Only one of the two short arms was rewarded with a sucrose pellet (the rewarded side was counterbalanced across mice but constant within a mouse) and the task for the mice was to learn which arm was rewarded. Each trial started with placing the mouse into the start section of the long arm. The guillotine door was then opened, and the mouse could choose one of the two short arms in a maximum of three minutes. After the mouse had entered a short arm with all four legs, the guillotine door of this arm was closed, and the mouse had time to find and eat the pellet (if it was the rewarded arm) or to explore the arm. The mouse was then put back to the start section, and about 5 s later, the next trial started. After finishing the 10 trials, the mouse was placed back into the home cage, the T-maze was cleaned, and the next mouse was tested. In the second phase of the experiment, reversal learning was tested. The procedure was identical to the acquisition phase; however, the contingency was reversed, i.e., the other short arm was rewarded. Importantly, on the fifth day of both phases (A5, R5), an 11th trial was added for which the T-maze was turned by 180 degrees. This trial was used to determine the learning strategy of the mice (allocentric vs. egocentric strategy).

In our second experiment, mice received nasal administration of NPS. The mice were habituated to the procedure of nasal administration (two administrations of 10 µL saline) during the habituation phase. During the acquisition and reversal phase, the mice received nasal administrations of either vehicle or NPS 2.5 h before their T-maze trials started.

### 4.7. Behavioral Procedure (Open Field)

The mice were placed into the open field arena and locomotor activity was recorded for one hour. They were then taken out of the open field and had either 10 µL vehicle or 1 mM NPS solution (counterbalanced across all animals) nasally applied. The mice were then put back into the open field, and locomotor activity was recorded for a further three hours. On the consecutive day, the mice were tested with the reversed treatment, i.e., those mice that received vehicle on the first day now received NPS, and vice versa.

### 4.8. Descriptive and Analytical Statistics

For the descriptive and statistical analysis of the data, GraphPad Prism 8 (GraphPad Software, La Jolla, CA, USA) and SYSTAT 13 (Systat Software, Inc., San Jose, CA, USA) were used. In addition to the number of correct choices, the switch cost ratio was also analyzed [71], i.e., the ratio of the mean performance during acquisition and during reversal. Data are shown in means ± SEM (standard error of the mean). Firstly, data were checked for normal distribution with the D’Agostino–Pearson test. Then, multifactorial analyses of variance (ANOVA) were performed accordingly. Proportions were analyzed with Pearson’s chi-square tests. For the analysis of reversal learning, animals that did not acquire the basic task (non-learners) were excluded. The criterion to be considered as a learner was an increase in correct choices within the acquisition phase of more than one. Data from one batch of mice (out of 14 batches) were excluded from the final analyses, since all mice from this batch did not lose weight with our usual amount of food restriction. These mice were very inactive throughout the experiment and were not motivated to look for the reward in the T-maze.

## 5. Conclusions

Our study shows that NPSR deficiency and nasal NPS administration have dissociative effects on discrimination learning and reversal in the T-maze. NPSR deficiency led to slightly improved acquisition of the task and a lower rate of non-learners but had no effects on reversal learning. Conversely, nasal application of NPS facilitated T-maze reversal learning and the use of an allocentric learning strategy without affecting the acquisition of the task or locomotor activity. Further studies are needed in order to understand the exact role of the NPS system in brain areas involved in the acquisition and reversal of discriminative T-maze learning as well as in strategy choice.

## Figures and Tables

**Figure 1 pharmaceuticals-14-00643-f001:**
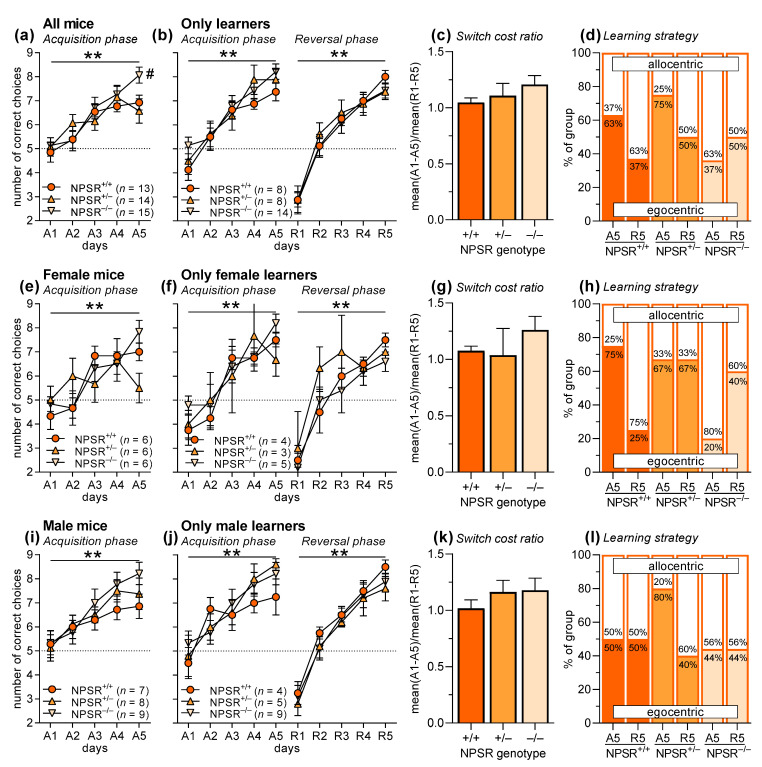
T-maze discrimination and reversal learning in wild-type, heterozygous and homozygous NPSR-deficient mice. Panels (**a**–**d**) depict the data for both sexes pooled, panels (**e**–**h**) and (**i**–**l**) show the data separated for female and male mice, respectively. (**a**) Mice learned well to discriminate the two arms of the T-maze; however, NPSR deficiency improved this learning. (**b**) For the analyses of reversal learning and learning strategy, mice that did not acquire discrimination learning during A1–A5 were excluded. It is notable that there were significantly less non-learners in NPSR^−/−^ mice. Mice that acquired discrimination learning (A1–A5) also learned the reversal task well (R1–R5), and NPSR deficiency had no effects. (**c**) This is confirmed by the analysis of the switch cost ratio, i.e., the ratio of the mean performance during acquisition and the mean performance during reversal learning. (**d**) In addition, learning strategy was not affected, either by phase of the experiment (acquisition, reversal) or by NPSR deficiency. Panels (**e**–**h**) and (**i**–**l**) show the same data separated by sex. Apart from the findings that female NPSR^+/−^ mice did not acquire the task (**e**), the result patterns are similar to those of the pooled data. ** *p* < 0.01, main effect of days (ANOVA), # *p* < 0.05, post-hoc comparison with NPSR^+/+^.

**Figure 2 pharmaceuticals-14-00643-f002:**
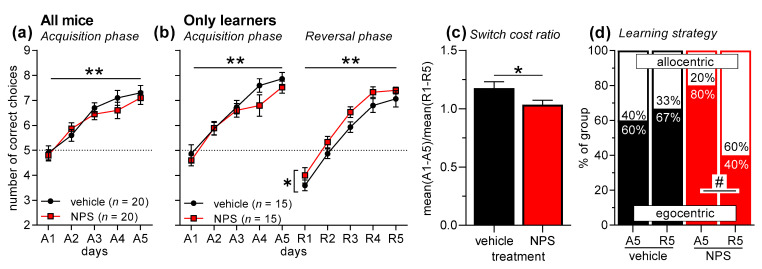
T-maze discrimination and reversal learning in male C57BL/6J mice after nasal NPS administration. (**a**) Mice learned to discriminate the two arms of the T-maze well; however, nasal NPS administration had no effects. (**b**) As before, mice that did not acquire discrimination learning during A1–A5 were excluded from the analyses of reversal learning and learning strategy. Mice that acquired discrimination learning (A1–A5) also learned the reversal task well (R1–R5). It is notable that nasal NPS administration improved reversal learning. (**c**) This NPS effect is confirmed by analyzing the switch cost ratio, i.e., the ratio of the mean performance during acquisition and the mean performance during reversal learning. (**d**) Vehicle-treated mice did not change their learning strategy from acquisition phase to reversal phase. However, NPS-treated mice showed a significant change from more egocentric learning in the acquisition phase to more allocentric learning in the reversal phase. ** *p* < 0.01, main effect of days; * *p* < 0.05, main effect of treatment (ANOVA or *t*-test, respectively); # *p* < 0.05 (chi-square test).

**Figure 3 pharmaceuticals-14-00643-f003:**
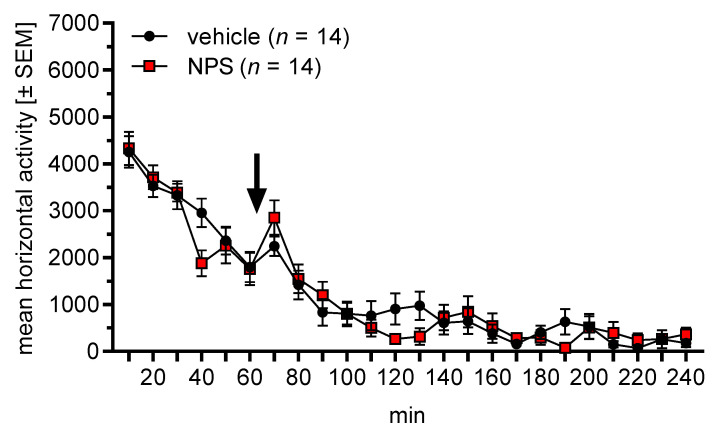
Effects of nasal NPS administration on locomotor activity (arbitrary units) in the open field. The arrow indicates the administration time point. Nasally applied NPS did not affect locomotor activity.

## Data Availability

Data are contained within the article.

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
