# Peer review of "Dissociative Effects of Neuropeptide S Receptor Deficiency and Nasal Neuropeptide S Administration on T-Maze Discrimination and Reversal Learning"

_pharmaceuticals, 2021, doi:10.3390/ph14070643_

Round 1
Reviewer 1 Report
Effects of Neuropeptide S Receptor Deficiency and Nasal Administration of Neuropeptide S on T-maze Discrimination and Reversal Learning
Comments for authors
Introduction:
The introduction should clarify the implication of the limbic system and particularly the implication of the hippocampus formation, in the cognitive flexibility. A larger summary (in a few lines) of what is cognition (or which part of what is cognition is investigated here) would enlarge the number of scientists interested by the paper; and citation level.
Another point that could rise up the global quality is the description of the molecular function of NPS, since its first function is not a synaptic function as classical neuromediators. Indeed, only cited that NPS effect on cognition has not been already addressed appears not sufficient for the quality of this paper. The cited literature gives information to link the already known localization, expression and molecular action of NPS and the goal of this study.
Results:
Why authors did not present the results (figure 1) with males separated from females. Actually, a significant difference appeared according the sex, but in the first figure results are pooled. Should be clarified since pooling males and females could lead to blind an effect.
I am surprised that authors did not use the strong effect of genotype in the cognitive outcome (learners versus non-learners). It seems that NPSR-/- mice present a higher rate of learning abilities than mice expressing NPSR. Does the NPSR expression act as an negative factor on learning performance? We see in addition in fig1a that NPSR-/- mice showed the highest level of performance (probably significant) at A5 point. This should also be clarified and discussed.
The number of animal used in the evaluation of locomotion (open field) is confusing. 8 and 6 animals are indicated in the text whereas n=14 is shown in figure 3. Also, it would be helpful for readers to know a little bit more about de "arbitrary unit" that was used to quantify the "horizontal activity". Please clarify and explain the quantified parameters.
The discussion appear quite relevant taking in account the strictly behavioral approach that gave only correlative conclusion. I a general point of view, it would be a great interest for this study to investigate the mechanistic role of NPS in the brain. Actually, histochemistry on brain slides, protein or mRNA quantification on brain sub-regions microdissections would up-grade the paper quality. This last remark is linked with the same point highlighted for the introduction.
Author Response
Introduction:
The introduction should clarify the implication of the limbic system and particularly the implication of the hippocampus formation, in the cognitive flexibility. A larger summary (in a few lines) of what is cognition (or which part of what is cognition is investigated here) would enlarge the number of scientists interested by the paper; and citation level.
ANSWER: We are grateful for these comments. In the revised manuscript, we now shortly described the implication of the limbic system, especially the hippocampus and the amygdala (lines 44-46) and explained cognitive flexibility better by giving an example for reversal learning (lines 35-38) which then also prepares the reader for the experimental protocol, we used.
Another point that could rise up the global quality is the description of the molecular function of NPS, since its first function is not a synaptic function as classical neuromediators. Indeed, only cited that NPS effect on cognition has not been already addressed appears not sufficient for the quality of this paper. The cited literature gives information to link the already known localization, expression and molecular action of NPS and the goal of this study.
ANSWER: We agree and added a statement why we think that the NPS system is well suited to modulate cognitive flexibility – based on NPSR expression and NPSR signaling (lines 64-67).
Results:
Why authors did not present the results (figure 1) with males separated from females. Actually, a significant difference appeared according the sex, but in the first figure results are pooled. Should be clarified since pooling males and females could lead to blind an effect.
ANSWER: We added now the data for female and male mice (new figure 1; lines 122-130, 140-142, 229-230). These data more or less confirmed the pooled data but also showed an acquisition deficit in female NPSR+/- mice.
I am surprised that authors did not use the strong effect of genotype in the cognitive outcome (learners versus non-learners). It seems that NPSR-/- mice present a higher rate of learning abilities than mice expressing NPSR. Does the NPSR expression act as an negative factor on learning performance? We see in addition in fig1a that NPSR-/- mice showed the highest level of performance (probably significant) at A5 point. This should also be clarified and discussed.
ANSWER: We now added a post-hoc test showing that NPSR-/- mice acquired the task better than NPSR+/+ mice (lines 99-101). Together with the higher responder rate in NPSR-/- mice, this suggests a better “cognitive outcome” in NPSR-/- mice. This enhanced performance is now also discussed (lines 221-224, 229-233, 237-239, 294-300)
The number of animal used in the evaluation of locomotion (open field) is confusing. 8 and 6 animals are indicated in the text whereas n=14 is shown in figure 3. Also, it would be helpful for readers to know a little bit more about de "arbitrary unit" that was used to quantify the "horizontal activity". Please clarify and explain the quantified parameters.
ANSWER: The animal number in the figure (n = 14) is the sum of eight C57Bl/6J and six NPSR+/+ mice used in this experiment. This is explained now in line 182.
According to the manufacturer, the units of the horizontal activity should be “cm”. We realized that this is by far not the case and therefore changed to “arbitrary units”.
The discussion appear quite relevant taking in account the strictly behavioral approach that gave only correlative conclusion. I a general point of view, it would be a great interest for this study to investigate the mechanistic role of NPS in the brain. Actually, histochemistry on brain slides, protein or mRNA quantification on brain sub-regions microdissections would up-grade the paper quality. This last remark is linked with the same point highlighted for the introduction.
ANSWER: We totally agree that it would be very nice to have additional molecular and/or histochemical data (e.g., combination of cFOS and NPSR). However, we are currently not able to perform such studies. For example, several (partly published) NPSR antibodies were not specific in our hands. We hope to be able to perform some follow-up studies in future going in the suggested directions.
Reviewer 2 Report
Overall, manuscript Pharmaceuticals-1232938 is a well-written and clear manuscript detailing the role and effects that NPS may have on reversal learning. There is some interesting data in the manuscript, all be the effects very mild and only just on the side of statistical significance. I have a couple of major comments with the manuscript, and a series of minor comments outlined below.
Major comments
Line 22: The abstract concludes with the statement “Taken together, the present data indicate that the NPS system is a modulator of cognitive flexibility”. I find this very hard to get behind when half of the date - the data from the knockouts - suggests the exact opposite. In fact, as the authors acknowledge in the discussion, the data from the knockouts suggests “beneficial effects of NPSR-deficiency” (line 193). Put another way, it suggests that endogenous NPS negatively regulates learning, which is largely in contrast to the authors’ original hypothesis. I believe this point needs to be addressed more specifically and the authors acknowledge that the data only partially supports a positive action of NPS, at least exogenous NPS, on cognitive flexibility and even the possibility of a negative effect on endogenous NPS.
General: I am wary of studies that use intranasal administration, particularly peptides, as I have concerns that the pharmacological site of action may be well away from that intended. It seems to me that the authors were in the ideal position to check the selectivity of the effect of endogenous NPS administration by running a control study administering NPS in the NPSR knockout. The manuscript would be very much improved by such a study as it would greatly increase confidence in the robustness and specificity of the effect. I lieu of such data the authors should at least acknowledge that there are some inconsistencies in their results that do not appear to agree with the original hypothesis.
Minor comments
Line 2: It would be better if the title made a more explanatory statement than simply stating “Effects of….” though I understand that the mixed bag of results may make this difficult.
Line 61: “side effects” would be better described as “confounding effects”
Line 64: Did the mutant mice differ in locomotor activity?
Line 75: I assume that all the environmental cues stayed in the same place when the maze was rotated. Is this the case? If so, please state.
Line 96: The difference between male and female mice is described as a “trend” as the p value was 0.06. Furthermore, the authors should state what the trend was rather than simply stating that there was a trend. However in line 185 the authors describe the difference as being significant.
Line 110: the double asterisk to indicate the level of statistical significance does not match that stated in the text around line 82.
Lines 165, 211: I think it is important to note that the effect was weak albeit statistically significant. I have a problem accepting this as robust, especially without evidence of the experiment being repeated and the authors obtaining the same results.
Line 188: Following on from my major comment above, some discussion of the fact that NPSR KO mice appeared to be less impaired (which I believe is contrary to their hypothesis) needs to be made.
Line 232: I think it a dangerous assumption to base their predictions on the penetrability of other neuropeptides.
General: Could the genetic background of the mice (B6.129S6 vs C57BL) explain any of the discrepancies in the findings?
General: The possibility of compensatory changes in the knockouts should be acknowledged.
Author Response
Overall, manuscript Pharmaceuticals-1232938 is a well-written and clear manuscript detailing the role and effects that NPS may have on reversal learning. There is some interesting data in the manuscript, all be the effects very mild and only just on the side of statistical significance. I have a couple of major comments with the manuscript, and a series of minor comments outlined below.
Major comments
Line 22: The abstract concludes with the statement “Taken together, the present data indicate that the NPS system is a modulator of cognitive flexibility”. I find this very hard to get behind when half of the date - the data from the knockouts - suggests the exact opposite. In fact, as the authors acknowledge in the discussion, the data from the knockouts suggests “beneficial effects of NPSR-deficiency” (line 193). Put another way, it suggests that endogenous NPS negatively regulates learning, which is largely in contrast to the authors’ original hypothesis. I believe this point needs to be addressed more specifically and the authors acknowledge that the data only partially supports a positive action of NPS, at least exogenous NPS, on cognitive flexibility and even the possibility of a negative effect on endogenous NPS.
ANSWER: Also based on the comments of the other reviewers and on additional statistical tests, we now emphasize the “beneficial effects” in the NPSR-/- mice more throughout the manuscript. Therefore, the respective part of the abstract is changed (lines 19-20, 23-27). However, we would like to emphasize that discrimination learning and reversal learning are two different cognitive ‘challenges’. Only the latter is related to cognitive flexibility and here, NPS had positive effects – which is in agreement with our working hypothesis. Regarding discriminative learning, we did not expect these beneficial effects of NPSR deficiency. These two effects indicate different roles of the NPS system in the brain areas involved for discrimination learning and cognitive flexibility, respectively (cf. lines 25-27, 298-304).
General: I am wary of studies that use intranasal administration, particularly peptides, as I have concerns that the pharmacological site of action may be well away from that intended. It seems to me that the authors were in the ideal position to check the selectivity of the effect of endogenous NPS administration by running a control study administering NPS in the NPSR knockout. The manuscript would be very much improved by such a study as it would greatly increase confidence in the robustness and specificity of the effect. I lieu of such data the authors should at least acknowledge that there are some inconsistencies in their results that do not appear to agree with the original hypothesis.
ANSWER: We accept this wariness, however, there are well-controlled studies in animals and humans showing that nasal administration of peptides can induce behavioral changes. In animals, there also a number of studies demonstrating that the peptides enter the brain and reach many brain regions. We agree that it would be a nice experiment to test nasal administration in NPSR-deficient mice. Of course, there might be the option that NPS has other pharmacological targets than NPSR (which could be proved by testing it in NPSR-deficient mice), however, we don’t understand why this should only be the case for nasal NPS and not for icv NPS.
We now acknowledge further potential ‘control experiments’ as well as the issue of the specificity of the effects in the last section of the discussion (lines 306-308).
Minor comments
Line 2: It would be better if the title made a more explanatory statement than simply stating “Effects of….” though I understand that the mixed bag of results may make this difficult.
ANSWER: We slightly modified the title (lines 2-3), however, it’s impossible to describe the different results in detail with just a few words.
Line 61: “side effects” would be better described as “confounding effects”
ANSWER: This was changed (line 73).
Line 64: Did the mutant mice differ in locomotor activity?
ANSWER: No (please see line 192, as well as Fig. 3 in Fendt et al., 2011)
Line 75: I assume that all the environmental cues stayed in the same place when the maze was rotated. Is this the case? If so, please state.
ANSWER: The environmental cues stayed on the same place which is now mentioned (line 88).
Line 96: The difference between male and female mice is described as a “trend” as the p value was 0.06. Furthermore, the authors should state what the trend was rather than simply stating that there was a trend. However in line 185 the authors describe the difference as being significant.
ANSWER: The nature of the trend is now explained in lines 111. The statement in the discussion (now line 224) refers to the first analysis of the T-maze experiment with all animals included (now line 97). The trend (now line 112) was observed after excluding the non-learners, which was done for the analyses of reversal learning and the learning strategy.
Line 110: the double asterisk to indicate the level of statistical significance does not match that stated in the text around line 82.
ANSWER: In general, we try to avoid using to many asterisks in the figures. ** P < 0.01 indicates a highly significant effect (is there more than highly significant?) and includes also Ps < 0.0001
Lines 165, 211: I think it is important to note that the effect was weak albeit statistically significant. I have a problem accepting this as robust, especially without evidence of the experiment being repeated and the authors obtaining the same results.
ANSWER: We deleted “robust” and changed the statement according to the reviewer’s suggestion (line 253).
Line 188: Following on from my major comment above, some discussion of the fact that NPSR KO mice appeared to be less impaired (which I believe is contrary to their hypothesis) needs to be made.
ANSWER: Please see our answer to the first two comments of this reviewer. In the discussion, now also the effects in NPSR-/- mice are discussed (lines 221-224, 228-232, 294-296, 298-300).
Line 232: I think it a dangerous assumption to base their predictions on the penetrability of other neuropeptides.
ANSWER: We agree and changed the respective statement (line 283).
General: Could the genetic background of the mice (B6.129S6 vs C57BL) explain any of the discrepancies in the findings?
ANSWER: We cannot totally exclude this.
However, the genetic background (C57BL/6J) of our NPSR breeding line (originally created in 129S6/SvEvTac) was confirmed by genome scanning already ca. 10 years ago. Meanwhile, the line was backcrossed ca. 10x more to C57BL/6.
General: The possibility of compensatory changes in the knockouts should be acknowledged.
ANSWER: Compensatory changes are now acknowledged (lines 237-239).
Reviewer 3 Report
In this study, the authors aimed to the role of the neuropeptide S (NPS) system in cognitive flexibility. The authors investigate the effect of NPS on T-maze discrimination and reversal learning in mice deficient of NPS receptors. NPSR-deficiency did not affect the performance in T-maze discrimination and reversal learning but was associated with a lower rate of non-learners. Furthermore, the authors examined whether nasal administration of NPS affect the T-maze discrimination, reversal learning, and the locomotor activity in wilt-type mice C67BL6/J. Nasal administration of NPS improved reversal learning and induced a shift to an allocentric learning strategy without affecting the acquisition of the task and locomotor activity.
The evidence and structure provided are good and will be useful for the field to have, however, there are some points that require authors’ attention.
- Differences in results between NPS receptors-deficient mice and administration of NPS
Nasal administration of NPS improved reversal learning and induced a shift to an allocentric learning strategy (Figure 2), while NPSR-deficiency did not affect the performance in the reversal learning (Figure 3). Why is the difference of the results? Please add the consideration of the differences to the section of discussion.
- Comparison with intracerebroventricular administration of NPS
In the past reports, NPS administration has the enhance effect of inhibitory avoidance learning, memory-enhancing effect and rescue memory deficits. Most of these studies are using intracerebroventricular administration of NPS. Why do you choose the nasal administration in the present study? In addition, nasal administration of NPS did not affect the acquisition of the task in T-maze discrimination. I suggest to the authors add intracerebroventricular administration of NPS in T-maze discrimination and reversal learning.
Author Response
In this study, the authors aimed to the role of the neuropeptide S (NPS) system in cognitive flexibility. The authors investigate the effect of NPS on T-maze discrimination and reversal learning in mice deficient of NPS receptors. NPSR-deficiency did not affect the performance in T-maze discrimination and reversal learning but was associated with a lower rate of non-learners. Furthermore, the authors examined whether nasal administration of NPS affect the T-maze discrimination, reversal learning, and the locomotor activity in wilt-type mice C67BL6/J. Nasal administration of NPS improved reversal learning and induced a shift to an allocentric learning strategy without affecting the acquisition of the task and locomotor activity.
The evidence and structure provided are good and will be useful for the field to have, however, there are some points that require authors’ attention.
- Differences in results between NPS receptors-deficient mice and administration of NPS
Nasal administration of NPS improved reversal learning and induced a shift to an allocentric learning strategy (Figure 2), while NPSR-deficiency did not affect the performance in the reversal learning (Figure 3). Why is the difference of the results? Please add the consideration of the differences to the section of discussion.
ANSWER: This point was also raised by the other reviewers. This is now addressed by different changes in the discussion of the revised manuscript (lines 221-224, 237-239, 294-296, 298-304).
- Comparison with intracerebroventricular administration of NPS
In the past reports, NPS administration has the enhance effect of inhibitory avoidance learning, memory-enhancing effect and rescue memory deficits. Most of these studies are using intracerebroventricular administration of NPS. Why do you choose the nasal administration in the present study? In addition, nasal administration of NPS did not affect the acquisition of the task in T-maze discrimination. I suggest to the authors add intracerebroventricular administration of NPS in T-maze discrimination and reversal learning.
ANSWER: We totally agree that testing icv administration of NPS in T-maze learning would be a very useful experiment. For the current study, nasal administration route was chosen due to the following advantages: it’s easier to learn and to perform (the study was performed by undergraduates), it’s less invasive (3R), and it’s a translational method (can be used in humans).
Since there is only 10 days time to revise the manuscript, we are not able to add any further experiments.
Reviewer 4 Report
This paper investigated the role of Neuropeptide S (NPS) ant its cognate receptor NPSR in cognitive flexibility. The model used is reversal learning assessed in a T-maze test. This model was used with mice lacking NPSR (NPSR-/-) compared with their wild type counterpart (NPSR+/+). Furthermore it was also tested if nasal administration of NPS affected the outcome of the T-maze test and locomotion. It was also assessed the learning strategy of the animals (allocentric vs. egocentric). The authors found no differences between (NPSR-/-) and (NPSR+/+) in T-maze suggesting that the NPSR deficiency does not affect reversal learning. Nasal administration of NPS slightly ameliorated the reversal learning with no effect on locomotion. Nasally-aministered NPS also promoted an allocentric learning strategy.
The paper is interesting and methods and results are presented in a clear way. NPS/NPSR system regulate many physiological processes including learning and memory but its effects on reversal learning and cognitive flexibility were not studied before. Thus this paper cover a novel angle of NPS/NPSR neurobiology.
I have a few comments:
The authors report that nasal administered NPS cause a shift in the mice learning strategy from egocentric to allocentric during the reversal phase (figure 2C). However it is not reported if there are significant differences between vehicle and NPS-administered mice in the acquisition and reversal phase. In the reversal phase 60% of NPS treated mice adopt the allocentric strategy while it was only 33% for the vehicle animals (figure 2C). Similarly, in the acquisition phase the allocentric strategy is adopted by 40% of vehicle mice and 20% of NPS mice (figure 2C). The differences seems potentially significant in both cases and interesting. Please comment. On the other hand if these differences exist this results would contrast with the fact that nasally administered NPS does not reach the basal ganglia that are involved in learning as specified in the discussion.
I have found interesting the analysis of learning strategy of the animals and how this is influenced by the NPS. I suggest to discuss the significance of the two strategies (allocentric vs. egocentric) and the implications of a change from one to another.
The absence of effects on T-maze by NPSR-/- is not totally unexpected. For instance, a lack of phenotype of NPSR-/- was previously seen with anxiety by Ruzza et al (2012). In many behavioural test, as also specified in the discussion, the NPSR-/- show only minor differences compared with their wild type counterpart. Thus since the evidences show that NPS/NPSR system is involved in many physiological processes one hypothesis is that some adaptations occur in NPSR-/- to compensate for the absence of functions of this important system.
Figure 1 (a-b): I suggest to change shapes not just the colours to the data point in these graphs. This would facilitate the reading when the paper is printed in B&W.
Author Response
This paper investigated the role of Neuropeptide S (NPS) ant its cognate receptor NPSR in cognitive flexibility. The model used is reversal learning assessed in a T-maze test. This model was used with mice lacking NPSR (NPSR-/-) compared with their wild type counterpart (NPSR+/+). Furthermore it was also tested if nasal administration of NPS affected the outcome of the T-maze test and locomotion. It was also assessed the learning strategy of the animals (allocentric vs. egocentric). The authors found no differences between (NPSR-/-) and (NPSR+/+) in T-maze suggesting that the NPSR deficiency does not affect reversal learning. Nasal administration of NPS slightly ameliorated the reversal learning with no effect on locomotion. Nasally-aministered NPS also promoted an allocentric learning strategy.
The paper is interesting and methods and results are presented in a clear way. NPS/NPSR system regulate many physiological processes including learning and memory but its effects on reversal learning and cognitive flexibility were not studied before. Thus this paper cover a novel angle of NPS/NPSR neurobiology.
I have a few comments:
The authors report that nasal administered NPS cause a shift in the mice learning strategy from egocentric to allocentric during the reversal phase (figure 2C). However it is not reported if there are significant differences between vehicle and NPS-administered mice in the acquisition and reversal phase.
ANSWER: We are not sure whether we understand this point of the reviewer. Of course, we reported whether there are significant differences between vehicle and NPS-administered mice in acquisition and reversal phase: NPS administration had no effects on acquisition (line 155; p = 0.55) but on reversal phase (line 162; p = 0.02). We now added a further analysis showing the switch cost ratio, i.e. the ratio of performance during acquisition and during reversal (see Figure 1 and 2; lines 116-119, 163-166, 374-376). This additional analysis confirmed the NPS effect, we observed on reversal learning (lines 163-165).
In the reversal phase 60% of NPS treated mice adopt the allocentric strategy while it was only 33% for the vehicle animals (figure 2C). Similarly, in the acquisition phase the allocentric strategy is adopted by 40% of vehicle mice and 20% of NPS mice (figure 2C). The differences seems potentially significant in both cases and interesting. Please comment. On the other hand if these differences exist this results would contrast with the fact that nasally administered NPS does not reach the basal ganglia that are involved in learning as specified in the discussion.
ANSWER: There were no effects of NPS administration on the single phases of the experiment. We now added these analyses (lines 169-170).
I have found interesting the analysis of learning strategy of the animals and how this is influenced by the NPS. I suggest to discuss the significance of the two strategies (allocentric vs. egocentric) and the implications of a change from one to another.
ANSWER: We agree that this is an interesting finding and expanded the discussion of this finding (lines 255-263, but see also 284-289).
The absence of effects on T-maze by NPSR-/- is not totally unexpected. For instance, a lack of phenotype of NPSR-/- was previously seen with anxiety by Ruzza et al (2012). In many behavioural test, as also specified in the discussion, the NPSR-/- show only minor differences compared with their wild type counterpart. Thus since the evidences show that NPS/NPSR system is involved in many physiological processes one hypothesis is that some adaptations occur in NPSR-/- to compensate for the absence of functions of this important system.
ANSWER: Actually, there are some effects of NPSR-deficiency, since these mice perform better and have an increased proportion of “learners” (lines 99-101, 105-107).
Figure 1 (a-b): I suggest to change shapes not just the colours to the data point in these graphs. This would facilitate the reading when the paper is printed in B&W.
ANSWER: Thanks for this suggestion. We changed the symbols in Figure 1.
Round 2
Reviewer 3 Report
This is the corrected version of the manuscript entitled "Dissociative effects of neuropeptide S receptor deficiency and nasal neuropeptide S administration on T-maze discrimination and reversal learning".
It is highly appreciated that authors did an effort to improve the manuscript and follow recommendations.
I can recommend considering this manuscript for publication.